# Possible role of caffeine in nicotine use onset among early adolescents: Evidence from the Young Mountaineer Health Study Cohort

**Alfgeir L. Kristjansson**[1,2]*, **Steven M. Kogan**[3], **Michael J. Mann**[4], **Megan L. Smith**[4], **Christa L. Lilly**[5], **Jack E. James**[6]

**1** Department of Social and Behavioral Sciences, West Virginia University School of Public Health, Morgantown, WV, United States of America, **2** Icelandic Center for Social Research and Analysis, Reykjavik, Iceland, **3** Department of Human Development and Family Science, University of Georgia College of Family and Consumer Sciences, Athens, GA, United States of America, **4** Department of Community and Environmental Health, Boise State University, College of Health Sciences, Boise, ID, United States of America, **5** Department of Epidemiology and Biostatistics, West Virginia University School of Public Health, Morgantown, WV, United States of America, **6** Department of Psychology, Reykjavik University, Reykjavik, Iceland

* alkristjansson@hsc.wvu.edu

## Abstract

### Background

Preventing nicotine use onset among children and youth is an important public health goal. One possible contributor that has received little empirical investigation is caffeine use. The goal of this study was to examine the possible contribution of caffeine to nicotine onset during early adolescence.

### Methods

We used data from the Young Mountaineer Health Study Cohort. Survey data were collected from 1,349 (response rate: 80.7%) 6th grade students (mean age at baseline 11.5 years) in 20 middle schools in West Virginia during the fall of 2020 and spring of 2021. We limited our analyses to students reporting never having used any form of nicotine at baseline. Logistic regression was employed in analyses.

### Results

Approximately 8% of participants reported having used nicotine at least once between baseline and the follow-up, and 4.7% reported solely using electronic nicotine delivery systems (ENDS) and no other forms of nicotine. In multivariable analyses, we controlled for many environmental, social, and behavioral variables known to influence nicotine use such as alcohol use, peer substance use, and perceived access to nicotine. We formulated our main independent variable, caffeine consumption, as continuous deciles. Any nicotine use, as well as ENDS use only at follow-up, were modeled as dependent variables. Caffeine was significantly associated with nicotine use in both models with ORs of 1.15 (1.04–1.27) and 1.13 (1.00–1.28).

**Data Availability Statement:** All relevant data are within the manuscript and its Supporting Information files.

**Funding:** Research reported in this publication was supported National Institutes Alcohol Abuse and Alcoholism of the National Institutes of Health under award number R01AA027241-01A1 (Kristjansson). The content is solely the responsibility of the authors and does not necessarily represent the official views of the National Institutes of Health. The funders had no role in study design, data collection and analysis, decision to publish, or preparation of the manuscript.

**Competing interests:** The authors have declared that no competing interests exist.

## Conclusions

Caffeine consumption among 6th grade non-nicotine users was associated with nicotine use at approximately 6-months follow-up.

## Introduction

Caffeine is the most widely consumed psychoactive substance in history [1]. Today, the majority of youth are regular consumers of caffeine-containing products, including soda, coffee, tea, candy, and increasingly, energy drinks, with many caffeinated products being directly marketed towards minors [2–4]. A recent review suggested that as many as 75% of children and adolescents aged 5–17 consume caffeine regularly [2]. Once ingested, caffeine is readily distributed throughout the body, exerting pharmacological actions at diverse sites, both centrally and peripherally [1]. Caffeine's main mechanism of action is competitive blockade of the neuromodulator adenosine, with A1 and A2A receptors appearing to be primary targets [5, 6]. In addition to influencing functionally important interactions between adenosine and dopamine receptors, caffeine effects include increased neurotransmitter activity in the CNS (anti-somnolent effect), constriction of cerebral and coronary blood vessels, renal diuresis, respiratory bronchodilation, increased secretion of gastrointestinal acid, and increased secretion of the catecholamine stress hormones epinephrine and norepinephrine [5, 7].

Whereas most published research on caffeine concerns adults, studies are increasingly being directed at younger age groups [2]. For young adolescents, the onset of caffeine consumption may be uniquely detrimental due to ongoing neurodevelopmental maturation coupled with increased educational and social demands. An area of special concern is the role caffeine may have in promoting early onset of other substance use, including nicotine. Caffeine's ability to stimulate dopamine release, a property shared with nicotine [8], suggests that caffeine may have the potential to augment the reinforcing and psychostimulant effects of nicotine. That possibility is all the more salient given that early nicotine use onset predicts later tobacco use [9] with younger age of onset strongly predicting future dependence [10]. For example, compared to onset at 14 years or older, first tobacco use at $\leq 13$ years of age forecasts both current and daily cigarette smoking and other tobacco product use, as well as the development of nicotine dependence [11]. Early nicotine onset also predicts the development of various negative developmental outcomes among adolescents and emerging adults such as criminal behavior [12], poor academic achievement [13], and increased alcohol and other drug use [14].

Studies of older adolescents (e.g., high-school students) and young adults have revealed associations between caffeine consumption and smoking behavior, including both cigarettes and use of electronic nicotine delivery systems (ENDS) such as e-cigarettes and/or vaping pens [15–17]. Two recent studies suggest that caffeine use by younger adolescents may also promote smoking behavior. First, a longitudinal study with middle school students [15] employed cross-lagged path models to prospectively test the association between total caffeine use and both cigarette smoking and ENDS use while controlling for reverse causation. Caffeine use was prospectively associated with both cigarette smoking and ENDS use but the reverse was not true. However, nicotine onset was not tested specifically in that analysis. Second, a cross-sectional study [18] found a positive association between caffeine consumption and nicotine use while controlling for demographic/background variables, although a temporal association could not be investigated in that analysis. Furthermore, neither of those studies [15, 18]

controlled for well-established covariates of nicotine use such as prior alcohol use, parental monitoring, perceived parental disapproval of substance use, and peer nicotine or substance use.

Considering the important role of dopamine in mediating the reward value of psychoactive drugs [19] and the fact that caffeine and nicotine both influence dopaminergic function, a connection between the two substances is biologically plausible. Research has repeatedly documented that early consumption of one habit forming substance tends to be linked with experimentation with and/or regular use of other substances, which may be caused by common biological and/or behavioral processes [10, 11, 20, 21]. The prevalence of caffeine consumption suggests that it may carry a uniquely heavy burden in the introduction of nicotine use among youth. This is especially relevant for new modes of nicotine ingestion such as via ENDS, which are increasingly prevalent [15]. However, to date, the prospective association between caffeine use and nicotine onset in young adolescents has not been tested empirically. Hence, the aim of the present study is to examine longitudinal associations between daily caffeine use and nicotine use onset among early adolescents while controlling for well-established social and behavioral covariates of nicotine use. We hypothesized that caffeine use at baseline would predict a greater likelihood of nicotine onset at follow-up.

## Materials and methods

### Sample and participants

The present analyses are based on the first two waves of survey data from the Young Mountaineer Health Study (YMHS) cohort. A single cohort of students enrolled in 20 geographically diverse public middle-schools in five counties in West Virginia are being followed twice per year from grades 6 through 8. Employing definition from the National Center for Educational Statistics, the schools include remote rural (1), distant rural (3), fringe rural (2), distant town (3), fringe town (1), small suburb (6), and small city (4) [16]. During the baseline assessment, from October—December 2020; 1,671 students attended regular school in either face-to-face or hybrid (part in person, part virtual) formats. Not included were participants who attended a virtual-only format, which in West Virginia was available only briefly as a limited option during the height of the COVID-19 pandemic. Of the students attending some amount of in-person school, 1,349 (80.7%) completed the study survey (mean age at baseline: 11.5 years). A follow-up survey was conducted during April and May of 2021 using identical data collection protocols. At baseline, 1,187 participants reported never using any type of nicotine product in their lifetime. Those represent the sample for our analyses.

### Procedure

The YMHS research team consists of the investigative team, study manager, three county data collection leaders (DCLs), and 20 supervising contact agents (SCAs, one in each school), to organize all data collection efforts. All five county Superintendents and twenty school principals approved participation in the study. Survey data collection was confidential, and protocols ensured anonymity of participants. No identifying information was collected. An introductory letter and/or email was sent to all parents and caregivers in September of 2020 to notify them about the study and provide an opportunity to opt their children out of participation and to put forth any questions or concern about the study. Consent procedures where explained and participants informed about their rights to deny participation without any repercussions. Data collection procedures utilize an honest-broker system to link individual data across study waves which secured individual confidentiality. Students responded to a confidential computer-based survey using Qualtrics software. The initial screen of the survey included a

consent description and explained that continuing the survey indicated verbal consent to participate. Data collection was supervised by research staff. Students responded to the surveys either inside schools or during designated online classes from home with their camera turned on, depending on accessibility during COVID-19 mitigation efforts. The Institutional Review Board of West Virginia University approved all study protocols (#1903499093A001).

## Measures

**Dependent variables.** *Nicotine use at follow-up*. Any form of nicotine use at follow-up was assessed with the following three questions: "In your lifetime, how many times have you smoked cigarettes (smoked a whole cigarette not just taken a puff)?"; "In your lifetime, how many times have you used e-cigarettes or vaping devices?" (i.e., ENDS); and "In your lifetime, how many times have you used other forms of tobacco that are not cigarettes or e-cigarettes (for example hookah, snuff, chewing tobacco)?" All questions were scored on a scale ranging from 1 = "Never" to 7 = "40 times or more often". For the purpose of our analyses, we recoded the responses into two dichotomized variables. First, we coded any use at follow-up for any of the three types of nicotine variables (or combination thereof) as "1", and no use ("never" to all three questions) as "0." Second, we coded those solely reporting ENDS use at follow-up as "1", and those who reported "never" for all other use were coded as 0. The second outcome variable therefore omits youth who reported using any other form of nicotine than ENDS at follow-up.

**Independent variables.** *Caffeine consumption*. The caffeine measure was designed to assess daily consumption from multiple types of beverages. This inventory has been previously validated in numerous publications [15, 22]. Respondents were asked, "How many cups/glasses/cans/or bottles do you usually drink every day of the following drinks?": Coffee, Tea, Caffeinated soda (e.g., Cola drinks, Mountain Dew, Dr. Pepper), and Energy drinks that contain caffeine (e.g., Red Bull, Monster, Rockstar, Bolt, etc). Response options ranged from 1 = "None" to 7 = "6 glasses/cups/cans/bottles or more". Participants were also asked: "How many caffeine "shots" (e.g., 5-Hour Energy) do you usually have each day?" and scored on the same scale as the previous four caffeine beverage questions. Each type of caffeine beverage was then weighted to reflect approximate proportional differences in caffeine content as reported previously [20, 27–29]: coffee (6x), tea (3x), soda/pop (1x), energy drinks (3x), and caffeine shots (12x). A continuous variable was created by summing the weighted responses for all five caffeine questions and converting them into an estimation of milligrams per day. Due to the wide range (0 – 3000mg) and high negative skew, the continuous variable was recoded into deciles for simpler interpretation.

## Control variables

In order to investigate the independent contribution of caffeine use, we controlled for factors associated with both nicotine onset and caffeine use including disrupted family structure [23], perceived parental disapproval/reactions to smoking and/or nicotine use [24], peer nicotine use and/or other substance use [25], perceived access to tobacco [26], parental tobacco use [27], and participant's alcohol use [28]. *Demographic covariates*: Participant gender was assessed with the question: "How do you describe your gender?", and the following four categories: 1 = "Boy", 2 = "Girl", 3 = "Gender Non-conforming", 4 = "Other (Please specify)". Due to low frequencies, groups 3 and 4 were merged as "other gender". Family structure was assessed with a multiple-response string of questions pertaining to "Who lives in your household". For the purpose of this analysis, this variable was coded into four levels with 1 = "Does not live with either biological parent", 2 = "Lives with biological mother", 3 = "Lives with

biological father", and 4 = "Lives with both biological parents". Given the low numbers of non-white students in WV, race was coded with 1 = "White" and "Other".

*Social and behavioral covariates.* Perceived parental reactions to alcohol, tobacco and other drug use was assessed with five questions headed by: "How do you think your parents/caregivers would react if you did any of the following?" followed by the items: "Smoked cigarettes", "Got drunk", "Used cannabis (marijuana)", "Used E-cigarettes or other vaping devises" or "Used snuff, chewing tobacco or other tobacco inserted in mouth". Responses ranged from 1 = "Totally against" to 4 = "They would not care". Scores were summed to form a scale (α = .88). Peer substance use was assessed with six questions headed by: "How many of your friends?", followed by the items: "Smoke cigarettes", "Use E-cigarettes or other vaping devices", "Drink Alcohol", "Have become drunk", "Use marijuana or other cannabis substances", or "Use other drugs". Response options ranged from 1 = "None" to 5 = "Almost all". Scores were summed to form a scale (α = .84). Perceived access to nicotine was assessed with two questions headed with: "How easy or hard would it be for you to get the following drugs if you wanted to:", followed by the items: "Cigarettes" and "E-cigarettes". Responses ranged from 1 = "Very difficult" to 4 = "Very easy". Responses were summed to form a scale ($r$ = .60). Parental tobacco use was assessed with the following questions: "Do any of the following people use tobacco on a daily basis:", followed by the items: "Mother/Stepmother" and "Father/Stepfather". Any use by either mother or father was coded as 1. Participant alcohol use was assessed with a single question: "In your lifetime, how many times have you had a drink of alcohol of any kind, even just a few sips (e.g. beer, wine, spirits, shots)?" Responses ranged from 1 = "Never" to 7 = "40 times +". For the purpose of our analyses, we recoded this variable with 0 = "Never", and 1 = "Yes, once or more" (13.2%).

Table 1 includes descriptive statistics for all study variables.

## Analyses

Our analyses were limited to participants who reported never using any type of nicotine product at baseline. We began our analyses by assessing the frequency of any nicotine use at follow-up. We then assessed which type(s) of nicotine product(s) participants reported using at follow-up. Finally, we ran two multivariable logistic regression models. Both models included caffeine recoded into deciles and inserted into the model as a continuous variable. Model 1 included any form of nicotine as a dichotomized outcome variable, and Model 2 included ENDS use only as the dependent variable (i.e., those never-users at baseline who reported only having used ENDS at follow-up). Covariates were included in the model as factors if appropriate for the variable type (e.g., gender was categorized into three levels, with "girl" as the reference group).

## Missing data analysis

The full data set included 16.7% missing values in our primary dependent variable, any nicotine use at follow-up. We examined whether this missingness was related to baseline demographic and predictor variables, including caffeine, gender, race, family structure group, baseline alcohol use, and parental smoking. Missing related to family structure ($p$ = 0.005) and baseline parental smoking ($p$ = 0.0002) with more missing data coming from respondents with at least one parent being a tobacco user and from respondents who did not live with both biological parents. However, both family structure and parental smoking were unrelated the outcomes in our multivariable analyses and were ultimately dropped from the analyses for model parsimony.

**Table 1. Descriptive statistics for all study variables (n = 1187).**

| Continuous Variables | Mean (SD) | Range |
|---|---|---|
| Average caffeine consumption in mg/day at baseline | 191.81 (301.64) | 0–3000 |
| Perceived parental reactions to alcohol, tobacco, and other drug use (n = 970) | 5.65 (1.94) | 5–20 |
| Peer substance use (n = 988) | 6.70 (2.04) | 6–30 |
| Perceived access to nicotine (n = 962) | 2.89 (1.61) | 2–8 |
| **Categorical Variables** | **N (%)** | |
| Any nicotine use at follow-up | | |
| Yes | 80 (8.09%) | |
| No | 909 (91.91%) | |
| Missing | 198 | |
| ENDS use only at follow-up | | |
| Yes | 46 (4.66%) | |
| No | 941 (95.34%) | |
| Missing | 200 | |
| Gender | | |
| Girls | 528 (46.11%) | |
| Boys | 593 (51.79%) | |
| Other | 24 (2.10%) | |
| Missing | 42 | |
| Race | | |
| White | 1038 (87.45%) | |
| Other | 149 (2.55%) | |
| Family Structure | | |
| Does not live with biological parent | 142 (11.96%) | |
| Lives with biological mother | 362 (30.50%) | |
| Lives with biological father | 78 (6.57%) | |
| Lives with both biological parents | 605 (50.97%) | |
| Any alcohol use at baseline | | |
| Yes | 156 (13.19%) | |
| No | 1027 (86.81%) | |
| Missing | 4 | |
| Daily caffeine use at baseline* | | |
| Coffee | 335 (28.27%) | |
| Tea | 603 (50.84%) | |
| Caffeinated soda | 824 (69.48%) | |
| Energy drinks | 111 (9.37%) | |
| Caffeine "shots" | 113 (9.56%) | |
| No caffeine use | 172 (14.49%) | |
| Parental smoking at baseline | | |
| Yes | 424 (36.21%) | |
| No | 747 (63.79%) | |
| Missing | 16 | |
| Among those who used any tobacco product at follow-up (n = 80), | | |
| what did they use?* | | |
| Smoked cigarettes | 22 (27.50%) | |
| E-cigarette/vaping | 66 (83.54%) | |
| Other products | 16 (20.00%) | |

*Multiple selections permitted

## Results

As shown in Table 1, 8.1% of participants had used at least one type of nicotine at follow-up and 4.7% reported having solely used ENDS and no other forms of nicotine. This means that around 58% of all new nicotine users at follow-up had solely used ENDS and no other forms of nicotine. Further multiple-response analyses revealed that around 83.5% of nicotine users at follow-up reported having used ENDS devices at least once, 27.5% had smoked cigarettes at least once, and 20.0% reported using some other form of nicotine. Regarding daily caffeine consumption, the highest daily prevalence was observed for caffeinated soda, ~69.5%, followed by Tea, ~51%.

The first model included any form of nicotine as the outcome. As shown in Table 2, for each decile point increase in caffeine use the odds of nicotine onset increased by 15%. Among the covariates the strongest contribution was observed for perceived parental reactions to alcohol, tobacco or other drug use (ATOD) use with OR = 1.19 (95% CI: 1.08–1.31) and peer substance use with OR = 1.18 (95% CI: 1.08–1.28). The second model included exclusive ENDS use as the outcome (dual users of ENDS and other forms of nicotine, or sole users of other forms of nicotine were excluded in this analysis). As shown in Table 2, for each point increase in caffeine use the odds of ENDS use onset increased by 13%. Among the covariates the strongest contribution was observed for perceived parental reaction to ATOD use with OR = 1.18 (95% CI: 1.06–1.32) and any alcohol use at baseline with OR = 2.41 (95% CI: 1.16–5.03).

## Discussion

In this study we analyzed two waves of survey data from a sample of 6th grade middle school students. Analyses were limited to students who reported having never used any type of nicotine product at baseline. At follow-up, approximately 8% of those students reported having used nicotine products at least once. The majority of those reported having used ENDS. Our primary question of interest was whether caffeine use at baseline was associated with nicotine use at follow-up. We modeled caffeine as a continuous variable by estimating mg/day and converting this variable into deciles in the analyses. We then modeled our outcome variables in two ways. First, as a composite measure of any form of nicotine use, including combustible cigarettes, ENDS, and both alternative tobacco (e.g., hookah) and smokeless tobacco. Second, given the popularity of ENDS as the primary mode of nicotine onset in young people [29], we

**Table 2. Logistic regression models with odds ratios and 95% Confidence Intervals, n = 926.**

| Variables | Model 1, DV: Any Nicotine Use at Follow-up | | | Model 2, DV: ENDS use only at Follow-up | | |
|---|---|---|---|---|---|---|
| | Exp (β) | 95% CI | P value | Exp (β) | 95% CI | P value |
| Caffeine mg/day (deciles) | 1.15 | 1.04, 1.27 | 0.003 | 1.13 | 1.00, 1.28 | 0.049 |
| **Covariates** | | | | | | |
| Gender (ref = Girls) | | | | | | |
| Boys | 0.98 | 0.55, 1.74 | 0.571 | 1.36 | 0.64, 2.87 | 0.790 |
| Other | 1.50 | 0.39, 5.66 | 0.526 | 2.33 | 0.52, 10.30 | 0.336 |
| Race (ref = Other) | | | | | | |
| White | 0.52 | 0.26, 1.04 | 0.064 | 0.99 | 0.37, 2.61 | 0.987 |
| Alcohol use at baseline (ref = No) | | | | | | |
| Yes | 1.33 | 0.69, 2.58 | 0.379 | 2.41 | 1.16, 5.03 | 0.018 |
| Perceived parental reactions to ATOD use | 1.19 | 1.08, 1.31 | 0.0005 | 1.18 | 1.06, 1.32 | 0.002 |
| Peer substance use | 1.18 | 1.08, 1.28 | 0.0001 | 1.08 | 0.97, 1.21 | 0.127 |
| Perceived access to nicotine | 1.38 | 1.08, 1.28 | <0.0001 | 1.18 | 1.06, 1.32 | 0.002 |

also ran our models to include only ENDS use as the outcome variable. Our models also included a selection of control variables known to contribute to caffeine use and nicotine onset.

Results indicated that caffeine consumption among non-nicotine using 6th grade students was positively related to the odds of nicotine use 6 months later. More specifically, in both Models 1 and 2, the continuous measure of caffeine suggests that for each decile unit increase in caffeine use the subsequent risk of nicotine use 6 months later increased by 13–15%. This means that those close to the top of the range were ~2.5 times more likely than non-users of caffeine to have used nicotine at least once between baseline and follow-up. Further, consistent with recent studies, ENDS use was observed to be the dominant form of nicotine onset in our sample of 6th grade students.

In the United States, as in many affluent countries, the introduction of ENDS was followed by a rapid increase in use. Introduced around 2007, by 2014 ENDS had become the most commonly used nicotine products among young people in the country [29]. After declaring ENDS use an epidemic among youth, the US Surgeon General [29] called for curbs on the marketing of ENDS to young people and the implementation of strategies to limit youth access to them. To date, ENDS use have remained the dominant form of nicotine uptake among young people [29]. The fact that ENDS use is currently much more common among youth than other forms of nicotine use has led to concerns about a reverse in the battle against nicotine addiction in the general population. The specific concern that use of ENDS may be contributing to increased numbers of nicotine users among young people is heightened by recent findings that the social and risk profiles of youth who smoke cigarettes do not differ markedly from those who use ENDS [30, 31]. Those developments underscore the importance of clarifying the role of other contributing factors, such as caffeine, in the onset and continued use of nicotine products, especially ENDS use.

Overall, our findings suggest that caffeine use is substantially associated with early onset of nicotine use. The mechanisms of action for this effect, however, have yet to be confirmed. Caffeine affects adenosine receptors which may affect striatal pathways associated with the production of dopamine and reward sensitivity. This area of the brain is particularly responsive during early adolescence [32]. For young adolescents, caffeine may be experienced as particularly rewarding, reinforcing the use of nicotine to regulate mood and the potential for nicotine addiction. The effects of caffeine may also be related to stressors associated with the transition from elementary to middle school, particularly within large public schools [33]. Youth experience increased academic demands and changes in school structure and routine. Additionally, media use and delayed sleep preferences among adolescents affect the likelihood of sleep deprivation which may affect youths' decisions to use both caffeine and nicotine. The finding that caffeine use was associated with both cigarette and ENDS use is additional suggestive confirmation of a biological basis for the widely reported association between caffeine and nicotine consumption.

Considering the ubiquity of caffeine [1], it is worth noting that caffeine consumption leads to physical dependence which is indicated by behavioral, physiological and subjective withdrawal effects (caffeine withdrawal syndrome). As few as eight hours of abstinence from regular intake may produce withdrawal symptoms, including sleepiness, lethargy, and headache [34, 35]. Referenced against standard criteria, principally those of the Diagnostic and Statistical Manual of Mental Disorders [36], caffeine has been labelled a "drug of abuse" [37]. Moderate consumption by youth has been found to be associated with diverse negative effects, including headache, nausea, drowsiness, fatigue, concentration difficulties, disrupted sleep, impaired academic achievement, mood disturbances, and aggressive behavior [22, 38, 39]. In addition to those effects, findings from the present study establish caffeine as a potentially important,

and until now largely unobserved, possible contributor to nicotine use in early adolescence. While not ignoring the plausibility of shared risk due to residual confounding, it is notable that previous analyses focused on frequency of nicotine use (rather than onset) that controlled for reverse causation suggest the link between caffeine and nicotine use among young adolescents likely is uni-directional [15]. Thus, increased arousal and alertness from caffeine's anti-somnolent effects coupled with caffeine's potential to augment the reinforcing and psychostimulant effects of nicotine may present a potent combination of effects for young adolescents coping with new and life challenges. Consequently, regular caffeine consumption, particularly during stressful periods of increased demands and responsibility typical of early adolescence, may contribute to the uptake and eventual regular use of nicotine.

Despite a number of strengths, our study also has some limitations. First, our sample is limited to 20 schools and five counties in West Virginia and may not be generalizable to other populations. Second, our sample was predominantly white/Caucasian. Third, our data relied solely on self-reports which are subject to recall bias. Fourth, our measurement of ENDS use did not distinguish between nicotine-bearing and non-nicotine bearing substances; neither did it preclude the possibility of cannabis use via ENDS. Future studies should be designed to assess such differences [40]. Fifth, although our measure of caffeine is comprehensive compared to most other studies of adolescent caffeine use, it did not include products known to include small amounts of caffeine, such as candy, chocolate, and yogurt. Finally, we did not collect data on where participants obtained the caffeine products they consumed (e.g., home or self-purchase). For example, outlets (e.g., stores and gas stations) from which caffeine products may have been purchased may also have nicotine advertisements and products on display. Exposure of that kind could be a factor in the consumption of nicotine products.

In conclusion, present findings add to growing evidence that early exposure to caffeine may increase the risk of early nicotine use. Rather than a single causal mechanism, multiple biological, behavioral, and social influences are likely to be involved. Both caffeine and nicotine influence dopaminergic function, which has an important role in determining the reward value of psychoactive drugs [20]. As such, early exposure to caffeine may serve to prime biological mechanisms that enhance the habit-forming potential of nicotine. In that context, the recent advent of highly concentrated caffeine products (e.g., caffeine "shots") commonly marketed directly at youth, should be of particular concern. Unfortunately, the current popularity of caffeine may encourage complacency regarding its negative effects. Conversely, no one seriously questions the harm posed by nicotine-containing products [41]. Thus, confirmation that early caffeine exposure may promote subsequent nicotine use should give rise to concern and efforts to limit caffeine consumption among youth.

## Supporting information

**S1 Dataset.**
(CSV)

## Author Contributions

**Conceptualization:** Alfgeir L. Kristjansson, Steven M. Kogan, Jack E. James.

**Data curation:** Alfgeir L. Kristjansson, Steven M. Kogan, Michael J. Mann, Megan L. Smith, Christa L. Lilly.

**Formal analysis:** Alfgeir L. Kristjansson, Christa L. Lilly, Jack E. James.

**Funding acquisition:** Alfgeir L. Kristjansson, Steven M. Kogan, Michael J. Mann, Christa L. Lilly.

**Investigation:** Alfgeir L. Kristjansson, Steven M. Kogan, Megan L. Smith.

**Methodology:** Alfgeir L. Kristjansson, Steven M. Kogan, Michael J. Mann, Megan L. Smith, Christa L. Lilly, Jack E. James.

**Supervision:** Alfgeir L. Kristjansson, Steven M. Kogan, Michael J. Mann.

**Writing – original draft:** Alfgeir L. Kristjansson, Michael J. Mann, Christa L. Lilly, Jack E. James.

**Writing – review & editing:** Alfgeir L. Kristjansson, Steven M. Kogan, Michael J. Mann, Megan L. Smith, Christa L. Lilly, Jack E. James.

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
