## [Decision Letter · Decision Letter 0]

23 Nov 2022

PONE-D-22-23716Possible role of caffeine in nicotine use onset among early adolescents: Evidence from the Young Mountaineer Health Study CohortPLOS ONE

Dear Dr. Kristjansson,

Thank you for submitting your manuscript to PLOS ONE. After careful consideration, we feel that it has merit but does not fully meet PLOS ONE’s publication criteria as it currently stands. Therefore, we invite you to submit a revised version of the manuscript that addresses the points raised during the review process.

Your manuscript has been assessed by one peer-reviewer and their report is appended below. 

The reviewer comments that the manuscript would benefit from more consistent use of terminology, and that some parts of the manuscript could be strengthened by some additional detail and/or clarification. In addition, the reviewer comments that some of the reported results do not appear to add up and that further clarification regarding the missing data is required. Furthermore, the reviewer suggests that the statistical analysis and discussions section requires further work.

In addition to the reviewer's concerns, the editorial office noted that the current study appears similar to your 2022 Preventative Medicine article "Caffeine consumption and onset of alcohol use among early adolescents" (https://doi.org/10.1016/j.ypmed.2022.107208). Specifically, the methodology section of both articles suggests that the same data set may have been used for both articles. This related article was not declared in the cover letter for this manuscript. Please clarify whether the same data set was used for both studies, and if so, please provide a detailed justification as to why this data set was separated into two separate, smaller studies, as opposed to presented as one larger manuscript.

Could you please revise the manuscript to carefully address the concerns raised?

Please note that we have only been able to secure a single reviewer to assess your manuscript. We are issuing a decision on your manuscript at this point to prevent further delays in the evaluation of your manuscript. Please be aware that the editor who handles your revised manuscript might find it necessary to invite additional reviewers to assess this work once the revised manuscript is submitted. However, we will aim to proceed on the basis of this single review if possible. 

We look forward to receiving your revised manuscript.

Kind regards,

Maria Elisabeth Johanna Zalm, Ph.D

Editorial Office

PLOS ONE

Journal Requirements:

2. Please provide additional details regarding participant consent. In the ethics statement in the Methods and online submission information, please ensure that you have specified (1) whether consent was informed and (2) what type you obtained (for instance, written or verbal, and if verbal, how it was documented and witnessed). If your study included minors, state whether you obtained consent from parents or guardians. If the need for consent was waived by the ethics committee, please include this information

3.  You indicated that you had ethical approval for your study. In your Methods section, please ensure you have also stated whether you obtained consent from parents or guardians of the minors included in the study or whether the research ethics committee or IRB specifically waived the need for their consent."

“Research reported in this publication was supported National Institutes Alcohol Abuse and Alcoholism of the National Institutes of Health under award number R01AA027241-01A1 (Kristjansson). The content is solely the responsibility of the authors and does not necessarily represent the official views of the National Institutes of Health.”

Additional Editor Comments:

The editorial office noted that the current study appears similar to your 2022 Preventative Medicine article "Caffeine consumption and onset of alcohol use among early adolescents" (https://doi.org/10.1016/j.ypmed.2022.107208). Specifically, the methodology section of both articles suggests that the same data set may have been used for both articles. This related article was not declared in the cover letter for this manuscript. Please clarify whether the same data set was used for both studies, and if so, please provide a detailed justification as to why this data set was separated into two separate, smaller studies, as opposed to presented as one larger manuscript, 

Reviewers' comments:

Reviewer's Responses to Questions

**Comments to the Author**

1. Is the manuscript technically sound, and do the data support the conclusions?

Reviewer #1: Yes

2. Has the statistical analysis been performed appropriately and rigorously? 

Reviewer #1: Yes

3. Have the authors made all data underlying the findings in their manuscript fully available?

Reviewer #1: No

4. Is the manuscript presented in an intelligible fashion and written in standard English?

Reviewer #1: Yes

5. Review Comments to the Author

Reviewer #1: This study utilized data from the Young Mountaineer Health Study Cohort to examine prospective associations between caffeine use and tobacco use at a 6-month follow-up among youth (mean age at baseline: 11.5 years).Strengths include a prospective cohort design that is able to capture new onset nicotine use, a relatively large analytic sample of youth (n=1187), and data are recent (2020-2021). This is overall a novel concept and worthy of publishing, but the manuscript needs substantial revisions.

Great job to authors! Specific comments below.

1. 2nd half of first paragraph in the Intro, starting at line 48 ("Caffeine's main mechanism of action...") - it would be helpful to add a sentence with a plain language summary of how these biological effects are relevant to the psychology of substance use. Consider how these effects may overlap with nicotine to prime the reader for the connection you will draw between the two substances.

2. Throughout the paper, authors use inconsistent terminology when referring to e-cigarettes, sometimes referring to them as e-cigs, sometimes as vaping, and sometimes as "e-cigarette and/or vaping" - choose one term, and just clarify in the Intro what you will call them

throughout the paper and stick to it.

3. In the "Sample and Participants" paragraph (page 5): "1,671 students attended school in either face-to-face or hybrid (part in person, part virtual) formats (i.e., not in virtual-only format) and thus were accessible to the study team." More details are needed here. Why were virtual-only students not accessible and how many were excluded? I also wonder whether schools in specific geographic regions (ex: remote rural or small city) were more or less likely to be virtual-only and thus be systematically excluded.

4. Did authors examine potential collinearity among the control variables? My concern is that the overlap in some of these variables may have led to overadjustment. Consider parsimony if there is little statistical rationale for keeping all of these covariates in the model.

5. In Table 1, Ns do not add up to total for most variables. For example, for the "Any nicotine use at follow-up" variable, n=80 were coded as Yes and n=909 were coded as No - these do not add up to the full N=1187. The same is true for: E-cigarette/Vaping only at follow-up, Gender, Any alcohol use at baseline, and Parental smoking at baseline. Please clarify whether these are missing data and add a footnote if so.

6. I think it would be worthwhile to show more info on types of caffeine consumed among the sample - what was the most common source? How many consumed more than one source? etc.

7. Did authors collect data on where participants were obtaining caffeine beverages from (e.g., from home, purchasing themselves)? If they were purchasing themselves from stores that have tobacco products on display, such as gas stations, it is possible that they also had greater exposure to tobacco advertisements.

8. Did authors collect data on duration of caffeine use? For example, if someone had been using caffeine regularly for a year vs. someone who had only been using for a couple months, would you expect their subsequent risk of nicotine use to differ?

9. Two regression models were run: 1) Any tobacco use vs. no tobacco use, and 2) Exclusive e-cig/vaping vs. no tobacco use. For Model 2, consider modeling the outcome as 3 levels: no nicotine use, vaping only, dual/poly use (i.e., vaping + some other nicotine product). This would help retain the sample size and would lend insight into whether there may be a dose-response relationship.

10. The Discussion could be strengthened by drawing more explicit connections between caffeine and nicotine - for example, an interesting point was raised that academic stressors increase from elementary to middle school (top of page 15), and the relevance of this point can be emphasized by showing that both substances can increase alertness and concentration, thus, both may be appealing/used in the same contexts to keep up with such academic stressors.

6. PLOS authors have the option to publish the peer review history of their article (what does this mean?). If published, this will include your full peer review and any attached files.

Reviewer #1: No

---

## [Author Response · Author response to Decision Letter 0]

26 Dec 2022

Manuscript PONE-D-22-23716

Editorial comments and our responses:

Comment 1: …the editorial office noted that the current study appears similar to your 2022 Preventative Medicine article "Caffeine consumption and onset of alcohol use among early adolescents" (https://doi.org/10.1016/j.ypmed.2022.107208). Specifically, the methodology section of both articles suggests that the same data set may have been used for both articles. This related article was not declared in the cover letter for this manuscript. Please clarify whether the same data set was used for both studies, and if so, please provide a detailed justification as to why this data set was separated into two separate, smaller studies, as opposed to presented as one larger manuscript. 

Our Response: Thank you for this observation. Both the article that is being considered by PLOS One and the Preventive Medicine article are indeed from the same data set. These data are from on the Young Mountaineer Health Study (YMHS), a cohort study funded by NIAAA where we are following a single cohort of middle school students over three years. The study plan enables us to test multiple hypotheses, about nicotine/tobacco use, alcohol use, and other drug use, with multiple mediating and moderating variables and different analysis techniques. A protocol manuscript for the YMHS was recently published and is accessible here: https://www.researchprotocols.org/2022/8/e40451/PDF

The main reasons for two separate manuscripts about caffeine and alcohol/nicotine onset are both theoretical and practical. 

First, nicotine is generally categorized as a stimulant. Alcohol, on the other hand, is primarily categorized as a depressant, although it is regarded as having stimulating properties in the early stages of drinking session. As such, the two drugs have different biological profiles. In addition, there tend to be differences in usage pattern. Nicotine often serves as the onset substance for early adolescents (average age in our sample is 11.5 years at Time 1), which is possibly a function of its ready availability in various forms, of which e-cigarettes and/or other vaping devices are currently the most common. The general decline in youth nicotine use in industrialized countries during the last 20 years is now threatened by the surge in use of these new products. West Virginia, where the YMHS sample is drawn, has typically experienced higher than average use of nicotine compared to other US States. The potential prospective impact of caffeine use on nicotine onset among early adolescents has not been previously studied in detail and warrants special attention. Youth alcohol use has also declined in industrialized countries, particularly during the last 10 years, but early onset of use is a marker of later use. Use of alcohol by youth tends to be strongly influenced by social setting, especially in relation to partying. Conversely, nicotine tends to be used across a wider range of social settings. Practically, discussing the relations between caffeine and nicotine/alcohol in two separate manuscripts enabled us to form the storyline and statistical tests in accordance with the difference in impact and social patterns of usage of the two drugs. We also felt that one manuscript would not give us sufficient space to adequately address the respective specific relationships that caffeine has with the other two drugs. Accordingly, we decided to submit a full-length manuscript for caffeine > nicotine onset, and another short communication manuscript for caffeine > alcohol onset. 

Comment 2: Please ensure that your manuscript meets PLOS ONE's style requirements, including those for file naming. 

Our Response: We have been careful to follow all PLOS One style requirements, including those for file naming. 

Comment 3: Please provide additional details regarding participant consent. In the ethics statement in the Methods and online submission information, please ensure that you have specified (1) whether consent was informed and (2) what type you obtained (for instance, written or verbal, and if verbal, how it was documented and witnessed). If your study included minors, state whether you obtained consent from parents or guardians. If the need for consent was waived by the ethics committee, please include this information

Our Response: Thank you. We have added further discussion concerning consent and IRB approval in the manuscript and the online submission information. We created a partnership with schools where the surveys were administered as a school-sponsored activity with two purposes. One was to provide basic epidemiological information for school health planning by participating schools. A secondary use was for research purposes. For research purposes, data was de-identified by an honest broker, then shared with the research team. This meant that the data, collected during the course of normal school activities, could not be linked to individual participants by the research team. Thus, for research purposes the data was essentially anonymous. This process was coupled with passive parental/caregiver consent procedures. Parents/caregivers received a take-home letter and an email describing the study and allowing parents to “opt out” their children. Informed consent from student participants was obtained, however, via a consent script on the opening survey screen which noted that continuing past that screen equaled consent to participate. Further, both prior introduction to the study at each school and the parent/caregiver communication (take-home letter and email), noted the rights of students to voluntarily participate or not, without any repercussion. Per our IRB requirement, the opening online survey screen also noted that students were not required to answer all questions, and that skipping questions or stopping participation during the survey, or refusing participation would have no consequences for them. 

Comment 4: You indicated that you had ethical approval for your study. In your Methods section, please ensure you have also stated whether you obtained consent from parents or guardians of the minors included in the study or whether the research ethics committee or IRB specifically waived the need for their consent. 

Our Response: Please see response to #3 above. 

Comment 5: Thank you for stating the following financial disclosure:

“Research reported in this publication was supported National Institutes Alcohol Abuse and Alcoholism of the National Institutes of Health under award number R01AA027241-01A1 (Kristjansson). The content is solely the responsibility of the authors and does not necessarily represent the official views of the National Institutes of Health.”

Our Response: Thank you. The additional sentence has been added to the funding statement which now reads: “Research reported in this publication was supported National Institutes Alcohol Abuse and Alcoholism of the National Institutes of Health under award number R01AA027241-01A1 (Kristjansson). The content is solely the responsibility of the authors and does not necessarily represent the official views of the National Institutes of Health. The funders had no role in study design, data collection and analysis, decision to publish, or preparation of the manuscript.”

Comment 6: Please include this amended Role of Funder statement in your cover letter; we will change the online submission form on your behalf.

Our Response: Thank you. The Role of Funder has been added to the cover letter (Our Response to Comment 5). 

Comment 7: In your Data Availability statement, you have not specified where the minimal data set underlying the results described in your manuscript can be found. PLOS defines a study's minimal data set as the underlying data used to reach the conclusions drawn in the manuscript and any additional data required to replicate the reported study findings in their entirety. All PLOS journals require that the minimal data set be made fully available. For more information about our data policy, please see http://journals.plos.org/plosone/s/data-availability.

Our Response: We will abide by PLOS rules for data access and availability and submit the data files for publication as Supporting Information. 

Reviewer 1 comments and our responses: 

Comment 1: 2nd half of first paragraph in the Intro, starting at line 48 ("Caffeine's main mechanism of action...") - it would be helpful to add a sentence with a plain language summary of how these biological effects are relevant to the psychology of substance use. Consider how these effects may overlap with nicotine to prime the reader for the connection you will draw between the two substances.

Our Response: As requested, new text has been added to the paragraph immediately following the paragraph highlighted by the reviewer, stating: “. . . caffeine’s ability to stimulate dopamine release, a property shared with nicotine [8], suggests that caffeine may have the potential to augment the reinforcing and psychostimulant effects of nicotine. That possibility is all the more salient given that . . .”

Comment 2: Throughout the paper, authors use inconsistent terminology when referring to e-cigarettes, sometimes referring to them as e-cigs, sometimes as vaping, and sometimes as "e-cigarette and/or vaping" - choose one term, and just clarify in the Intro what you will call them

throughout the paper and stick to it.

Our Response: These discrepancies have been remedied. Throughout the narrative we now refer to all e-cigarette use or vaping of nicotine as use of “electronic nicotine delivery systems” or “ENDS”. This is clarified in the Introduction section. 

Comment 3: In the "Sample and Participants" paragraph (page 5): "1,671 students attended school in either face-to-face or hybrid (part in person, part virtual) formats (i.e., not in virtual-only format) and thus were accessible to the study team." More details are needed here. Why were virtual-only students not accessible and how many were excluded? I also wonder whether schools in specific geographic regions (ex: remote rural or small city) were more or less likely to be virtual-only and thus be systematically excluded.

Our Response: During the height of the COVID-19 pandemic and before vaccines became available, particularly in the fall of 2020 (vaccines did not become publicly available until early 2021), most WV counties operated their schools in three distinct ways (County Boards of Education represent the local authority for schools in each respective county): a) regular in-person school (albeit with many in-house restrictions), b) virtual from home, or c) a hybrid form of in-person and virtual. The latter option was chosen by many parents and students in the targeted counties. In addition, for parts of the fall semester in 2020, schools were closed several times, based on both local and state-wide COVID-19 exposure measures that were updated weekly, and decided on by WV State Education Authorities and local Boards of Education. A fourth option for schooling was also run solely by the WV State. This option utilized a separate curriculum, was virtual only, and unrelated to county school curriculum. Families and children that selected this option for schooling in the fall of 2020 became ineligible for the study, because our sampling frame and all study plans and preparations were based on collaboration with County Boards of Education in the five participating counties. Limited data exists on the group that participated in the state-wide virtual school program, but local insights suggest that this group comprised less than 8% of the total student population and was characterized by more affluent students compared to the rest of the student population. Unfortunately, we had very limited opportunity to obtain data on this group and the option was closed after the 2020 – 2021 school calendar year. In addition, due to the nature of the pandemic many students also chose to come back to regular school at some point during the fall 2020 and spring 2021 semesters and did not spend a whole semester in the State virtual option. This WV State option is what we are referring to in the sentence sited by the reviewer. In an attempt to clarify this further in the manuscript without overly complicating the storyline we have revised the sentence to read as follows: 

“During the baseline assessment, from October - December 2020; 1,671 students attended regular school in either face-to-face or hybrid (part in person, part virtual) formats (i.e., not in virtual-only WV State format which was a limited option during the height of the COVID19 era) and thus were accessible to the study team. “

Comment 4: Did authors examine potential collinearity among the control variables? My concern is that the overlap in some of these variables may have led to overadjustment. Consider parsimony if there is little statistical rationale for keeping all of these covariates in the model.

Our Response: Thank you. We did assess collinearity and found the Variance Inflation Factor to be below 1.16 for all variables. We also re-ran all models, and for parsimony we omitted family structure and parental smoking from the final reported models as those were unrelated to the outcomes in both models reported in Table 2. 

Comment 5: In Table 1, Ns do not add up to total for most variables. For example, for the "Any nicotine use at follow-up" variable, n=80 were coded as Yes and n=909 were coded as No - these do not add up to the full N=1187. The same is true for: E-cigarette/Vaping only at follow-up, Gender, Any alcohol use at baseline, and Parental smoking at baseline. Please clarify whether these are missing data and add a footnote if so.

Our Response: The differences are due to variation in missing values. For clarification we have added sample sizes and missing cases to the table. 

Comment 6: I think it would be worthwhile to show more info on types of caffeine consumed among the sample - what was the most common source? How many consumed more than one source? etc.

Our Response: Frequency of reported daily caffeine use by type has been added to Table 1, as well as the proportion of non-users. Those details are also discussed briefly in the Results section. 

Comment 7: Did authors collect data on where participants were obtaining caffeine beverages from (e.g., from home, purchasing themselves)? If they were purchasing themselves from stores that have tobacco products on display, such as gas stations, it is possible that they also had greater exposure to tobacco advertisements.

Our Response: Unfortunately, we did not. However, the point is well taken, leading us to include the omission as a limitation in the penultimate paragraph of the Discussion. 

Comment 8: Two regression models were run: 1) Any tobacco use vs. no tobacco use, and 2) Exclusive e-cig/vaping vs. no tobacco use. For Model 2, consider modeling the outcome as 3 levels: no nicotine use, vaping only, dual/poly use (i.e., vaping + some other nicotine product). This would help retain the sample size and would lend insight into whether there may be a dose-response relationship.

Our Response: Thank you. As suggested, we conducted this analysis by running a multinomial logistic regression model (see Table below). The findings are consistent with our original analyses. The multinomial model, however, resulted in a more complicated storyline that affected the clarity of the manuscript. We thus elected to retain the original strategy of reporting on two binary logistic models. 

Table 3. Multinomial logistic regression models excluding tobacco users of substances other than vaping, with odds ratios and 95% Confidence Intervals, n = 915.

 Model 3: DV E-cig/Vaping only (n=42) and E-cig/Vaping with other substances (n=19) relative to no tobacco use (n=854)

Variables Est. SE P value

Intercept vaping only -7.76 0.73 <0.0001

Intercept vaping plus other substances -6.35 0.68 <0.0001

 Exp (β) 95% CI P value

Caffeine mg/day (deciles) 1.15 1.03, 1.27 0.008

Covariates 

Gender (ref=Girls)

 Boys

 Other 

1.35

1.14 

0.72, 2.53

0.23, 5.45 

0.597

0.980

Race (ref=Other)

 White 

0.36 

0.18, 0.72 

0.003

Alcohol use at baseline (ref=No)

 Yes 

1.41 

0.71, 2.82 

0.318

 Perceived parental reactions to ATOD use 1.17 1.06, 1.29 0.001

 Peer substance use 1.17 1.08, 1.27 0.0001

 Perceived access to nicotine 1.42 1.24, 1.63 <0.0001

Comment 9: The Discussion could be strengthened by drawing more explicit connections between caffeine and nicotine - for example, an interesting point was raised that academic stressors increase from elementary to middle school (top of page 15), and the relevance of this point can be emphasized by showing that both substances can increase alertness and concentration, thus, both may be appealing/used in the same contexts to keep up with such academic stressors.

Our Response: In response to the reviewer’s request, new text has been added to the Discussion, as follows: “Thus, increased arousal and alertness from caffeine’s anti-somnolent effects coupled with caffeine’s potential to augment the reinforcing and psychostimulant effects of nicotine may present a potent combination of effects for youth coping with new and increased life challenges. Consequently, regular caffeine consumption, particularly during stressful periods of increased demands and responsibility typical of early adolescence, may contribute to the uptake and eventual regular use of nicotine.”

---

## [Decision Letter · Decision Letter 1]

20 Feb 2023

PONE-D-22-23716R1Possible role of caffeine in nicotine use onset among early adolescents: Evidence from the Young Mountaineer Health Study CohortPLOS ONE

Dear Dr. Kristjansson,

Thank you for submitting your manuscript to PLOS ONE. After careful consideration, we feel that it has merit but does not fully meet PLOS ONE’s publication criteria as it currently stands. Therefore, we invite you to submit a revised version of the manuscript that addresses the points raised during the review process.

We look forward to receiving your revised manuscript.

Kind regards,

Tommaso Martino, M.D.

Academic Editor

PLOS ONE

Reviewers' comments:

Reviewer's Responses to Questions

**Comments to the Author**

1. If the authors have adequately addressed your comments raised in a previous round of review and you feel that this manuscript is now acceptable for publication, you may indicate that here to bypass the “Comments to the Author” section, enter your conflict of interest statement in the “Confidential to Editor” section, and submit your "Accept" recommendation.

Reviewer #1: All comments have been addressed

Reviewer #2: All comments have been addressed

Reviewer #3: All comments have been addressed

2. Is the manuscript technically sound, and do the data support the conclusions?

Reviewer #1: (No Response)

Reviewer #2: Yes

Reviewer #3: Yes

3. Has the statistical analysis been performed appropriately and rigorously? 

Reviewer #1: (No Response)

Reviewer #2: Yes

Reviewer #3: Yes

4. Have the authors made all data underlying the findings in their manuscript fully available?

Reviewer #1: (No Response)

Reviewer #2: Yes

Reviewer #3: Yes

5. Is the manuscript presented in an intelligible fashion and written in standard English?

Reviewer #1: (No Response)

Reviewer #2: No

Reviewer #3: Yes

6. Review Comments to the Author

Reviewer #1: (No Response)

Reviewer #2: Thank you for inviting me to review the revised manuscript Possible role of caffeine in nicotine use onset among early adolescents: Evidence from the Young Mountaineer Health Study Cohort for consideration in PLOS ONE. The authors have addressed all comments from the previous round of review. However, the authors can still improve on descripting their data collection procedure (see comments below). There are several minor grammatical-type inconsistencies throughout the paper. Since PLOS ONE does not copyedit accepted manuscripts, I would suggest the authors carefully read through the manuscript and make any needed small edits (see comments below for some suggested edits).

Introduction:

Line 68 change “(e.g. high-school students)” to (e.g., high-school students)

Materials and Methods

check throughout the manuscript for 1 vs 2 spaces after periods (see line 98, also line 27 in the abstract)

Line 104 “WV State”, just write out West Virginia

Line 105 change “COVID19” to “COVID-19” to be consistent throughout the paper

Table 1:

“ENDS only at follow-up” change variable to “ENDS use only at follow-up”

“Coffee 335 (28.27%” missing closing parentheses

Procedure:

Line 123-124 I am a bit confused about what “designated class-room hours from home” means. Did research staff come into individual homes? Earlier in the manuscript it sounded like all participants were in school at least part time and all others excluded. This phrase made me think maybe I misunderstood what was previously stated.

I don’t think this applies but in case it does: Did the authors check for any differences in responses between students who took the survey at home vs at school? I worry that students taking the survey at home where parents may be present could influence responses.

Results:

Line 212-213 You write “ENDS devices” in several places. Just say ENDS and remove the word devices. This is also in the discussion several times. Please check and change throughout the manuscript.

Line 225 “The second model included exclusive ENDS as...” (insert “use”)

Discussion:

Line 242 add the word “use” after ENDS “...we also ran our models to include only ENDS [use] as the outcome variable”

Line 249 “Further, consistent with recent studies, ENDS was” change to “ENDS use was” or make another change. The sentence reads oddly.

Line 253 “...commonly used nicotine product...” change to “...commonly used nicotine products...”

Reviewer #3: (No Response)

7. PLOS authors have the option to publish the peer review history of their article (what does this mean?). If published, this will include your full peer review and any attached files.

Reviewer #1: No

Reviewer #2: No

Reviewer #3: No

---

## [Author Response · Author response to Decision Letter 1]

23 Feb 2023

Manuscript PONE-D-22-23716R1

Reviewer 2, additional comments:

Comment 1: Thank you for inviting me to review the revised manuscript Possible role of caffeine in nicotine use onset among early adolescents: Evidence from the Young Mountaineer Health Study Cohort for consideration in PLOS ONE. The authors have addressed all comments from the previous round of review. However, the authors can still improve on descripting their data collection procedure (see comments below). There are several minor grammatical-type inconsistencies throughout the paper. Since PLOS ONE does not copyedit accepted manuscripts, I would suggest the authors carefully read through the manuscript and make any needed small edits (see comments below for some suggested edits). 

Our Response: Thank you for taking the time to provide additional comments to improve our work. We greatly appreciate the reviewer’s dedication in helping us provide a quality manuscript for further consideration. 

Comment 2: Introduction: Line 68 change “(e.g. high-school students)” to (e.g., high-school students). 

Our Response: The requested change has been made. 

Comments 3-5: Materials and Methods. 

Check throughout the manuscript for 1 vs 2 spaces after periods (see line 98, also line 27 in the abstract). 

Line 104 “WV State”, just write out West Virginia

Line 105 change “COVID19” to “COVID-19” to be consistent throughout the paper

Our Response: We have carefully checked all double spaces after periods and made the additional requested changes. 

Comments 6-7: Table 1: 

“ENDS only at follow-up” change variable to “ENDS use only at follow-up”

“Coffee 335 (28.27%” missing closing parentheses

Our Response: Thank you. Those changes have been made. 

Comments 8-9: Procedure: 

Line 123-124 I am a bit confused about what “designated class-room hours from home” means. Did research staff come into individual homes? Earlier in the manuscript it sounded like all participants were in school at least part time and all others excluded. This phrase made me think maybe I misunderstood what was previously stated.

I don’t think this applies but in case it does: Did the authors check for any differences in responses between students who took the survey at home vs at school? I worry that students taking the survey at home where parents may be present could influence responses. 

Our Response: We agree with the reviewer that accessing survey participants from home warrants special attention with regards to the potential of outside influences, such as from parents or caregivers. This is precisely what we did and attempted to convey in the narrative. During the COVID-19 pandemic many schools operated solely online for different periods of time. If our data collection was planned to take place while schools were operating solely online, we attempted to minimize potential interference from others (such as parents/caregivers or siblings) by ensuring that the surveys were run during the time of a designated class while the student camera was turned on, as opposed to simply sending a survey link to the student’s email address. This way we preserved a degree of supervision regarding who would be responding to the survey, and control over the time when the survey was responded to by each student. We have revised the sentence pertaining to this issue. It now reads: “Students responded to the surveys either inside schools or during designated online classes from home with their camera turned on, depending on accessibility during COVID-19 mitigation efforts”. 

Comments 10-13: Results/Discussion: 

Line 212-213 You write “ENDS devices” in several places. Just say ENDS and remove the word devices. This is also in the discussion several times. Please check and change throughout the manuscript.

Line 225 “The second model included exclusive ENDS as...” (insert “use”)

Line 242 add the word “use” after ENDS “...we also ran our models to include only ENDS [use] as the outcome variable”

Line 249 “Further, consistent with recent studies, ENDS was” change to “ENDS use was” or make another change. The sentence reads oddly.

Our Response: We have carefully reviewed and revised all instances of “ENDS” so that the manuscript now solely states “use of ENDS”, or “ENDS use” and the word “devices” has been removed. 

Comment 14: Line 253 “...commonly used nicotine product...” change to “...commonly used nicotine products...”

Our response: The change has been made.

---

## [Decision Letter · Decision Letter 2]

10 Apr 2023

PONE-D-22-23716R2Possible role of caffeine in nicotine use onset among early adolescents: Evidence from the Young Mountaineer Health Study CohortPLOS ONE

Dear Dr. Kristjansson,

Thank you for submitting your manuscript to PLOS ONE. After careful consideration, we feel that it has merit but does not fully meet PLOS ONE’s publication criteria as it currently stands. Therefore, we invite you to submit a revised version of the manuscript that addresses the points raised during the review process.

We look forward to receiving your revised manuscript.

Kind regards,

Tommaso Martino, M.D.

Academic Editor

PLOS ONE

Journal Requirements:

Additional Editor Comments:

As suggested by our referee, the health-related negative effects of nicotine are well known, also in its pure form. Please avoid such sentences.

Reviewers' comments:

Reviewer's Responses to Questions

**Comments to the Author**

1. If the authors have adequately addressed your comments raised in a previous round of review and you feel that this manuscript is now acceptable for publication, you may indicate that here to bypass the “Comments to the Author” section, enter your conflict of interest statement in the “Confidential to Editor” section, and submit your "Accept" recommendation.

Reviewer #2: All comments have been addressed

Reviewer #4: (No Response)

2. Is the manuscript technically sound, and do the data support the conclusions?

Reviewer #2: Yes

Reviewer #4: Yes

3. Has the statistical analysis been performed appropriately and rigorously? 

Reviewer #2: Yes

Reviewer #4: Yes

4. Have the authors made all data underlying the findings in their manuscript fully available?

Reviewer #2: Yes

Reviewer #4: Yes

5. Is the manuscript presented in an intelligible fashion and written in standard English?

Reviewer #2: Yes

Reviewer #4: Yes

6. Review Comments to the Author

Reviewer #2: I have no further comments regarding this manuscript. I appreciate the authors' edits and openness to feedback.

Reviewer #4: Please provide rationale for recoding caffeine intake into deciles instead of using raw values.

Statement at the end of discussion "Conversely, although nicotine products are not all equally harmful to health, no one seriously questions the potential for harm posed by nicotine-containing products in general" is quite an unsupported generalization and should be rephrased or removed. There are many well known and well studied harms of nicotine, independent of the health risks caused by the non-nicotine components of nicotine-containing products.

7. PLOS authors have the option to publish the peer review history of their article (what does this mean?). If published, this will include your full peer review and any attached files.

Reviewer #2: No

Reviewer #4: No

---

## [Author Response · Author response to Decision Letter 2]

15 Apr 2023

Manuscript PONE-D-22-23716R2

Reviewer 4, additional comments:

Comment 1, Editorial: Please review your reference list to ensure that it is complete and correct. If you have cited papers that have been retracted, please include the rationale for doing so in the manuscript text, or remove these references and replace them with relevant current references. Any changes to the reference list should be mentioned in the rebuttal letter that accompanies your revised manuscript. If you need to cite a retracted article, indicate the article’s retracted status in the References list and also include a citation and full reference for the retraction notice.

Our Response: We have carefully reviewed the reference list and made minor changes to four references. No reference was retracted or added. 

Comment 2: Please provide rationale for recoding caffeine intake into deciles instead of using raw values. 

Our Response: We have added the following sentence at the end of the description of the caffeine variable (lines 152-153): “Due to the wide range (0 – 3000mg) and high negative skew, the continuous variable was recoded into deciles for simpler interpretation.”

Comment 3 and Editorial: Statement at the end of discussion "Conversely, although nicotine products are not all equally harmful to health, no one seriously questions the potential for harm posed by nicotine-containing products in general" is quite an unsupported generalization and should be rephrased or removed. There are many well known and well studied harms of nicotine, independent of the health risks caused by the non-nicotine components of nicotine-containing products.

Editorial: As suggested by our referee, the health-related negative effects of nicotine are well known, also in its pure form. Please avoid such sentences.

Our Response: We have revised the sentence to read as follows: “Conversely, no one seriously questions the harm posed by nicotine-containing products [41]”.

---

## [Decision Letter · Decision Letter 3]

28 Apr 2023

Possible role of caffeine in nicotine use onset among early adolescents: Evidence from the Young Mountaineer Health Study Cohort

PONE-D-22-23716R3

Dear Dr. Kristjansson,

We’re pleased to inform you that your manuscript has been judged scientifically suitable for publication and will be formally accepted for publication once it meets all outstanding technical requirements.

Kind regards,

Tommaso Martino, M.D.

Academic Editor

PLOS ONE

Reviewers' comments:

Reviewer's Responses to Questions

**Comments to the Author**

1. If the authors have adequately addressed your comments raised in a previous round of review and you feel that this manuscript is now acceptable for publication, you may indicate that here to bypass the “Comments to the Author” section, enter your conflict of interest statement in the “Confidential to Editor” section, and submit your "Accept" recommendation.

Reviewer #4: All comments have been addressed

2. Is the manuscript technically sound, and do the data support the conclusions?

Reviewer #4: Yes

3. Has the statistical analysis been performed appropriately and rigorously? 

Reviewer #4: Yes

4. Have the authors made all data underlying the findings in their manuscript fully available?

Reviewer #4: Yes

5. Is the manuscript presented in an intelligible fashion and written in standard English?

Reviewer #4: Yes

6. Review Comments to the Author

Reviewer #4: (No Response)

7. PLOS authors have the option to publish the peer review history of their article (what does this mean?). If published, this will include your full peer review and any attached files.

Reviewer #4: No

---

## [Editor Report · Acceptance letter]

3 May 2023

PONE-D-22-23716R3 

Possible role of caffeine in nicotine use onset among early adolescents: Evidence from the Young Mountaineer Health Study Cohort 

Dear Dr. Kristjansson:

I'm pleased to inform you that your manuscript has been deemed suitable for publication in PLOS ONE. Congratulations! Your manuscript is now with our production department. 

Kind regards, 

on behalf of

Dr. Tommaso Martino 

Academic Editor

PLOS ONE